# Brindley’s Glands Volatilome of the Predator *Zelus renardii* Interacting with *Xylella* Vectors

**DOI:** 10.3390/insects14060520

**Published:** 2023-06-03

**Authors:** Ugo Picciotti, Miguel Valverde-Urrea, Francesca Garganese, Federico Lopez-Moya, Francisco Foubelo, Francesco Porcelli, Luis Vicente Lopez-Llorca

**Affiliations:** 1Dipartimento di Scienze del Suolo, della Pianta e degli Alimenti (DiSSPA), University of Bari Aldo Moro, 70125 Bari, Italy; ugo.picciotti@uniba.it (U.P.); francesco.porcelli@uniba.it (F.P.); 2Department of Marine Science and Applied Biology, Laboratory of Plant Pathology, University of Alicante, 03690 Alicante, Spain; mrvu1@gcloud.ua.es (M.V.-U.); lv.lopez@ua.es (L.V.L.-L.); 3Department of Organic Chemistry, Institute of Organic Synthesis, University of Alicante, 03690 Alicante, Spain; foubelo@ua.es

**Keywords:** *Harpactorini*, leafhopper assassin bug, antifragility, infochemicals, OQDS, CoDiRO

## Abstract

**Simple Summary:**

*Zelus renardii* is one of the alien insects recently acclimatised to the Mediterranean basin. The trivial name “leafhopper assassin bug” preludes the reduviid prey preference. In the Mediterranean, *Zelus renardii* adapted to preying on vicarious indigenous species, including *Philaenus spumarius* (Hemiptera: Aphrophoridae), the European vector of *Xylella fastidiosa*. Reduviidae has several thoracic glands, pair or unpair, whose secretions may regulate adult insect behaviours, such as defence, alarm, and mating. *Zelus renardii* also possesses a pair of Brindley’s glands, each consisting of about one hundred elements secerning in a reservoir with an outlet that opens at the thoracoabdominal limit. Stressful events elicit the production and secretion of a semiochemical bouquet, acting as alarm pheromones. This bouquet comprises 2-methyl-propanoic acid, 2-methyl-butanoic acid and 3-methyl-1-butanol as significant components, effectively repel conspecifics and suggest the role of Brindley’s glands as alarm pheromone foci. *Zelus renardii* reduces the production of alarm pheromones and the chance of being detected by prey interacting with *P. spumarius*. Alternatively, the alarm pheromone could help the predator to mark its territory, avoiding interaction with a conspecific. Evidence of the ability of *Philaenus spumarius* to perceive and react to the predator’s semiochemical would provide a further means to manage transmission and infection by *Xylella fastidiosa*.

**Abstract:**

Alien species must adapt to new biogeographical regions to acclimatise and survive. We consider a species to have become invasive if it establishes negative interactions after acclimatisation. *Xylella fastidiosa* Wells, Raju et al., 1986 (XF) represents Italy’s and Europe’s most recent biological invasion. In Apulia (southern Italy), the XF-encountered *Philaenus spumarius* L. 1758 (Spittlebugs, Hemiptera: Auchenorrhyncha) can acquire and transmit the bacterium to *Olea europaea* L., 1753. The management of XF invasion involves various transmission control means, including inundative biological control using *Zelus renardii* (ZR) Kolenati, 1856 (Hemiptera: Reduviidae). ZR is an alien stenophagous predator of *Xylella* vectors, recently entered from the Nearctic and acclimated in Europe. *Zelus* spp. can secrete semiochemicals during interactions with conspecifics and prey, including volatile organic compounds (VOCs) that elicit conspecific defence behavioural responses. Our study describes ZR Brindley’s glands, present in males and females of ZR, which can produce semiochemicals, eliciting conspecific behavioural responses. We scrutinised ZR secretion alone or interacting with *P. spumarius*. The ZR volatilome includes 2-methyl-propanoic acid, 2-methyl-butanoic acid, and 3-methyl-1-butanol, which are consistent for *Z. renardii* alone. Olfactometric tests show that these three VOCs, individually tested, generate an avoidance (alarm) response in *Z. renardii.* 3-Methyl-1-butanol elicited the highest significant repellence, followed by 2-methyl-butanoic and 2-methyl-propanoic acids. The concentrations of the VOCs of ZR decrease during the interaction with *P. spumarius*. We discuss the potential effects of VOC secretions on the interaction of *Z. renardii* with *P. spumarius*.

## 1. Introduction

Current global trade and the movement of plant material increase the introduction of alien species [1]. These species must overcome biotic and abiotic stresses to survive, colonise, and reproduce [2]. In this scenario, native species meet allochthonous ones, generating new interactions. These interactions can lead to mutualistic interplays (positive interactions) or biological invasions (negative interactions) [3]. Negative interactions can turn alien species into invasives, leading to infestations of plant pests or pathogen epidemics.

The quarantine bacterium *Xylella fastidiosa* Wells et al., 1987 (XF) subsp. *Pauca* ST53 [4,5] entered from Costa Rica to Italy (Apulia), developing the consequent epidemic on olive trees, showing how an alien species can invade new hitherto-unexplored territories. Indeed, XF-infected plants ultimately cease production and die within a few years [6]. In Apulia, XF encountered Aphrophoridae xylem-feeders (Spittlebugs, Hemiptera: Auchenorrhyncha), which acquire, transmit, and spread the bacterium in *Olea europaea* L., 1753 populations, a new host. Several Aphrophoridae can acquire and transmit XF. However, *Philaenus spumarius* L., 1758 (PS) is the main vector for population size and infective abilities [7]. The ability to transmit XF has changed PS’s status from a marginal insect to a key pest for olive trees in the Mediterranean Basin, leading to increased attention in managing PS infections [8]. The currently recommended strategy first suggests minimising PS populations during juvenile stages. Management then focuses on surviving adults, to prevent acquisition and transmission, or limiting infection events to one per plant adult to slow or to stop XF invasion. The aim is to kill vectors during their first feeding on olive trees, particularly in areas still free from the bacterium [7].

The basis of the infection management strategy is the integration of several control actions. This strategy involves resistant olive tree cultivars, mechanical and chemical control of PS juvenile instars, and chemical and biological control of PS adults [7,9,10,11,12,13,14]. The biological control of PS can involve the predator *Zelus renardii* Kolenati, 1856 (ZR, Hemiptera: Reduviidae) [12,14], recently acclimated in the Mediterranean Basin [14,15]. The reduviid will attack adult PS and other olive pests such as *Bactrocera oleae* (Rossi, 1790) (Diptera: Tephritidae) [14]. Furthermore, ZR could be a potential biological control agent for other invasive alien pests recently entering Europe, such as *Macrohomotoma gladiata* Kuwayama, 1908 (Hemiptera: Psyllidae) and *Drosophila suzukii* Matsumura, 1931 (Diptera: Drosophilidae) [16,17]. As a biocontrol agent, ZR provides new interactions among alien and native species. These novel interactions offer the opportunity to unveil the mechanisms of integrating alien species into pre-existing trophic networks in the biogeographical area of entrance. Therefore, analysing how they interact with their biotic environment is relevant.

Olfaction is the primary sense by which insects perceive single or semiochemical blends [18,19], which are carbon-based volatile organic compounds (VOCs) vaporising at 20 °C and 0.01 kPa [20]. VOCs can play the role of interspecific (pheromones) and intraspecific (allelochemicals) communication [19,21,22,23]. Semiochemicals serve as olfactory cues [18,24,25], eliciting behavioural or physiological replies. Cimicomorpha and Pentatomomorpha (Hemiptera: Heteroptera) possess peculiar metathoracic glands [26], whose secretions dictate several adult insect behaviours [27]. Many Reduviidae species have Brindley’s and metathoracic exocrine glands in pairs [27,28,29,30,31]. Secreted VOCs are involved in defence, alarm, and mating [29,31,32,33]. Brindley’s glands have orifices located dorsally in the metathorax [34,35] and secrete alarm pheromones in the event of disturbance of the reduviids [29]. Metathoracic glands have lateral/ventrolateral outlets and dedicated dispersive apparatuses dictating aggregation, copulation, and defence [35,36]. The Harpactorinae subfamily, including *Zelus* spp., lacks metathoracic glands [37].

*Zelus renardii* uses semiochemicals for partner or prey searching [29,30,38,39,40,41]. ZR also supplements olfactory information with visual cues when encountering mobile prey [14]. In contrast, vibrational communication predominates in PS. Nevertheless, PS also utilises intraspecific chemical communication systems [42]. However, little knowledge exists on the chemical communication of PS underlying intraspecific interactions. It is assumed that PS relies on vibrational rather than chemical communication to manage mating and other intraspecific interactions [43,44].

Therefore, this work focuses on ZR and PS VOCs, alone and in reciprocal interactions. We aim to establish background information to understand recognition among *Z. renardii* individuals and between ZR and its *P. spumarius* prey, the XF vector.

## 2. Materials and Methods

### 2.1. Insect Collection

ZR specimens for the study came from a September 2021–October 2022 collection in a *Citrus* CV orchard near Elche (38°14′54 N 0°41′43 W). ZR adults were collected using 60 mL single-use probes (Deltalab, Rubí, Spain) purposely perforated for ventilation. We obtained 18 ZR adults (males and females) for the study.

Adult PS thrived in a dicot-dominated field near Jijona (38°32′25 N 0°30′38 W) and were swept using a net from June to September 2022. PSs were transferred from the sweeping net into vented 5 mL microtubes (Deltalab, Rubí, Spain). We avoided mouth aspirator collection because abrupt draw-up would have inflicted low-pressure stresses on PS, eventually modifying the VOC Aphrophoridae profile.

### 2.2. Brindley’s Glands

To demonstrate that the Brindley’s glands secrete VOCs capable of eliciting behavioural responses, we scrutinised ten ZR adults (five males and five females). The insects originated from 2018–2021 collections given during a study on reduviid mass-breeding performance. Adults were allowed to rest in the dark until December 2022 and then fixed in 75% *v*/*v* EtOH/distilled water, prepared using pure bioethanol (PVG Liquids N.V., Gent, Belgium). We took ten ZR adults from those available in the collection of the Forensic Entomology Laboratory of DiSSPA (University of Bari Aldo Moro, Bari, Italy).

Studying the insects, we first obtained a general view from the right side of the mesothorax and metathorax plus part of the abdomen. Then, we attempted to observe the Brindley’s glands on each half of every individual with light microscopy. Then, we focused on SEM evidence.

Each ZR was passed from EtOH to distilled water for 12–48 h to extract the alcohol. Later, we cut away and preserved the legs at the trochanter, prothorax, and abdomen at the third urite. After one/two days of water replacement, gentle shaking, and rest in a vial on a warm plate (40–50 °C), we replaced the water with a fluid made by mixing 1/1 SDS (Sodium Dodecyl Sulphate in distilled water 50% *w*/*v*) and branded pure hand dishwashing soap. Porcelli [45] suggested a similar procedure, but we simplified the protocol by eliminating KOH, apart from the first two studied ZR specimens, which were also double stained [46]. The parts of each ZR were cleared first in soap and surfactant and then in Essig’s Aphid Fluid (EAF) on a warm plate until they showed the cuticular details of Brindley’s glands under a stereoscope. We cut each ZR part following the sagittal plane, obtaining two exposed Brindley’s glands. A Zeiss Phomi II and a Zeiss Tessovar, purposely modified for thick slide imaging and equipped with Olympus PEN cameras, were used to study and perform bright field macro and microscopy imaging.

We used the same ZR parts scrutinised by light microscopy for the SEM study. The parts underwent steps from EAF to 75% EtOH, 99% EtOH, and propyl-acetate [7]; the last step was run in glassware with a minute outlet for slow solvent evaporation. After propyl-acetate evaporation, we obtained well-dehydrated cuticle parts for observations. Each ZR half was inserted into a Ø 12.5 × 8 mm height alloy SEM stub (Agar Scientific Ltd., Stansted, UK) using double-sided conductive tape (Ted Pella, Inc., Redding, CA, USA) [47], exposing the insect cuticle’s internal surface. Insect parts were first imaged using a TM3000 Hitachi SEM in charge-up reduction mode. They were then lightly coated with gold/palladium sputtered by an Edwards S150 Ion Sputter Coater for 30″ and 8 mA current to observe more detail. The electron beam was accelerated at KV 5, 15, and 15 for analysis. In Cryo-SEM mode, the TM3000 was used with sublimated ultrapure water (Milli-Q^®^ Lab Water Solution, Sigma-Aldrich, St. Louis, MO, USA), embedding the ZR parts at –25/–50 °C to expose the cuticle [48]. Meshwork and other imaging details were performed at –45 °C in charge-up reduction mode. Adobe Photoshop^®^ (Adobe Incorporated, San Jose, CA, USA) was used to edit the images in grayscale and false colours.

### 2.3. Experimental Setup for VOC Detection

A Gas Chromatography–Mass Spectrometer was used to gather VOC data from one ZR, one PS, or one ZR + one PS per vial (HS, crimb, FB, 20 mL, clr, cert, 100 PK: Agilent Technologies, Santa Clara, CA, USA). We starved ZR and PS 24 h before their placement in vials at 23 °C under laboratory conditions.

The treatments targeted VOCs produced by a ZR or a PS alone in a vial or the VOCs from a net-limited interaction in the vial (Figure 1). A top circle (0.5 cm Ø) and a longitudinal net wall (2 × 0.5 cm) of polyurethane square mesh (1 × 1 mm) (Figure 1C) were used to limit the contact between the insects. An empty vial served as a control versus a vial with nets to discriminate an eventual “net effect”. We performed seven biological replicates per treatment (ZR, PS, and ZR + PS).

We also searched for VOCs produced upon PS aggregation. This test was comprised of four treatments (increasing the number of PS per vial), testing 18 PS in total (males and females) in trials with either 1, 2, 5, or 10 individuals placed in separate vials (Figure 2).

The solid phase microextraction (SPME) holder with a fused silica fibre (10 mm; Ø 80 µm; DVB/CWR/PDMS, Agilent Technologies, Santa Clara, CA, USA) absorbed VOCs by exposing the carrier’s fibre to the vials’ headspace for one hour at room temperature (approx. 23 °C).

### 2.4. Gas Chromatography–Mass Spectrometry (GC/MS) Analyses

The fibre placed into the GC injector (Agilent 5977B network mass spectrometer; Agilent model 7890B gas chromatograph; column: DB-624, length 30 m, 0.25 mm ID, 1.4 μm, Agilent) underwent a 4 min desorption at 250 °C in split/splitless mode. The fibre injector was a robot MPS (Multipurpose Sampler; Gerstel GmbH & Co. KG, Mülheim an der Ruhr, Germany). The fibre was conditioned at 250 °C for 10 min before being injected into each sample. The chromatographic program used was started at 40 °C for 5 min and then later was increased by 5 °C/min up to 230 °C to maintain the temperature for 10 min. In total, the analysis time was 53 min. The ionisation source value for the electronic impact was 70 eV at 230 °C, with a mass range between 25 and 450 amu. The detector was a simple quadrupole at 150 °C. The NIST11 library allowed for the tentative identification of VOCs. After each chromatographic run, the software generated a chromatogram and VOC list.

The VOCs obtained were clustered into major VOCs with a match ≥ 50% and a peak height ≥ 100.000 ppm, or minor VOCs with a match ≥ 50% and a peak height < 100.000 ppm.

### 2.5. Olfactometer Analysis

Four-arm PET (polyethene terephthalate) olfactometers were used to study the behavioural response to selected VOCs found in the volatilomes. Each olfactometer consisted of a central chamber, 12 cm in Ø and 6 cm tall, connected to four arms equally angled at 90°. Each arm was 3 cm in Ø and 6 cm long.

The lists of VOCs per treatment (ZR, PS, and ZR + PS) were compared using venn diagrams to determine which VOCs were found most frequently in the GC/MS analysis. We selected the VOCs in at least four out of seven replicates for the behavioural analysis.

Selected VOCs (2-methyl-propanoic acid, 2-methyl-butanoic acid, and 3-methyl-1-butanol) were tested separately in pouch dispensers (3.5 cm × 2.5 cm) of miracloth (Merck KgaA, Darmstadt, Germany). Each dispenser contained 2 g of 60 A silica gel (70–200 μ, Carlo Erba Reagents s.r.l., Cornaredo, Italy). We added 2 µL of the given pure compounds to the silica gel in each dispenser for the trials. One hundred and twenty tests (forty per compound) were performed to understand the effect of the selected VOCs on ZR behaviour.

We tested each VOC against four adult wild-collected ZR males. We placed each VOC randomly into two of the four peripheral chambers, while the other two were left empty as negative controls. The tests were set up by rotating the ZRs between four different olfactometers and letting them rest for at least five minutes between each test. Each ZR underwent olfactory stimulation for 30 min. We rinsed the olfactometer after each trial with n-hexane, ethanol, and distilled water and dried it with paper towels to remove any residual VOCs. Olfactometry was conducted under laboratory conditions (23 °C, approx. 60% HR) with a seasonal photoperiodic condition (approx. 10:14 L:D).

We assessed the number of times the predator approached the VOC and the negative control, recording the final ZR choice after half an hour. Forty olfactometer trials were run for each VOC.

### 2.6. Chemicals

Pure 2-methyl-propanoic acid and 2-methyl-butanoic acid were synthesised at the Institute of Organic Synthesis of the Department of Organic Chemistry at the University of Alicante (Spain), and the pure 3-methyl-1-butanol was from TCI (Tokyo Chemical Industry Co., Ltd., Tokyo, Japan).

### 2.7. Data Analysis

A multivariate generalised linear model (GLM) with a Poisson distribution of the error was conducted to examine changes in the set of volatilomes of ZR alone with the prey. Furthermore, a 999-permutation multivariate analysis of variance (PERMANOVA) was used to analyse the Bray–Curtis dissimilarity matrix. The treatment (ZR and ZR + PS) was a fixed factor. A betadisper was used to test the multivariate homogeneity of group dispersion. However, due to the lack of multivariate homogeneity of variances, we used the F-value modification [49]. A SIMPER analysis was used to determine which VOC contributed most to the differences. In addition, we conducted a univariate study of the variation produced in each VOC using a generalised linear mixed model (GLM) with a Gaussian family error distribution.

A GLMM with a binomial error distribution was used to determine the final choices of ZR to move towards or away from the VOC in the olfactometer tests.

A GLMM with Poisson distribution error was used to analyse the number of approaches to VOCs. The treatment (control and compound) and the experiment were the fixed factors.

Both models treated the individual identity of the insect as a random factor to control for the non-independence of repeated measures on the same individual.

All statistical analyses were performed using R version 4.1.2 (R Core Team, 2022) using “manyglm” from the “mvabund” package [50] for multivariate and univariate GLM models. PERMANOVA was used with the “adonis2” function of the “vegan” package [51]. Furthermore, the GLMM models [52] were used with the function “glmer” from the package “lme4”, while the function “simulateResiduals” from the package “DHARma” [53] was used to perform the model diagnosis.

## 3. Results

### 3.1. Brindley’s Glands Morphology

The right-side view focused on the adult ZR thorax and abdomen, showing relevant cuticular details in reflected light (Figure 3a). The marks indicated the pronotum (Figure 3a: 1), mesothoracic coxa (Figure 3a: 2), metathoracic coxa (Figure 3a: 3), second urite (Figure 3a: 4), third urite (Figure 3a: 5), first urite (Figure 3a: 8), hemelytron (Figure 3a: 9), mesopleuron (Figure 3a: 11), patchy cuticle areas (Figure 3a: 12), meshwork evaporatorium (Figure 3a: 13), and the Brindley’s gland reservoir outlet (Figure 3a: 15), serving to put into context the observations of Brindley’s glands and associated structures. Moreover, Figure 3 and Figure 4 show details from the outside, while Figure 5 and Figure 6 demonstrate the same areas and details from the inside of the insect. The same marks refer to the same details in all images. We scrutinised the same details, namely, the same parts from the same insect, using light and SEM microscopy, making the resulting description somewhat repetitive in numbering; a table of numbers/descriptions is provided to help follow the results (Table 1).

Macrography and SEM (Figure 3a,b) were conducted to identify the major cuticular elements: first (Figure 3: 7) and second (Figure 3: 6) abdominal spiracle; meshwork (Figure 3a,b and Figure 4a,b: 14); gutter (Figure 3a,b and Figure 4a,b: 16); abdominal finger (Figure 3a,b and Figure 4a,b: 21); and mesothoracic flap (Figure 3a and Figure 4b,d: 10). The external side of the cuticle did not show any other relevant details. Transmitted bright light microscopy showed the interior with one Brindley’s gland reservoir per side. Each reservoir (Figure 5a–c: 18) is immediately below the first abdominal spiracle and slightly above the second abdominal spiracle (Figure 3a,b and Figure 5a,c: 6, 7). The second (Figure 5a,c: 6) abdominal spiracle and meshwork (Figure 5a,c: 14) remained consistent with the external observations. In the last figures, minute drop-like sacculi (Figure 5a,c,e: 19) appeared to be associated and possibly connected with the reservoir. SEM showed the second abdominal spiracle (Figure 6: 6), the tracheal trunks and subdivision (Figure 6: 17a,b,c), and Brindley’s reservoir (Figure 6: 18) sheltered by sacculi (Figure 5d and Figure 6: 19) and partially hidden by tracheal subdivisions 17b and 17c (Figure 5d; 6).

The main tracheal trunk of the second spiracle was found to run to the metathorax, and its bifurcation stayed over the sagittal side of the gland (Figure 6). Reservoirs were membranous, flask-like, and slightly dorso–ventrally flattened (Figure 5a,b). The inflated reservoir measured approximately 0.12 × 0.10 mm, and no appreciable size differences were found between ZR males and females. Two bi-convex outlets corresponded to each meshwork-like metathoracic area (Figure 3a,b and Figure 5c).

A mantle of drop-like cuticular sacculi covered each reservoir (Figure 5c,d and Figure 6). We suggest that sacculi in ZR correspond to the B-type glandular units described by Barrett [54] in *Rodnius prolixus* Stål, 1859 (Reduviidae: Triatominae).

ZR glandular units were 5–6 µm in diameter (Figure 5d,e). In *R. prolixus*, the B-type sacculi joined Brindley’s reservoirs by proper ducts randomly distributed and oriented toward the reservoir’s internal surface. ZR detached sacculi show no duct and a single glandular unit type only. A-type secretory units with elongated, U-shaped sacculi [34,54] were not found in ZR.

Brindley’s gland reservoirs in *Z. renardii* connected in a short cuticular duct opening just above the supra-coxal lobe of the metathoracic pleurae, as described in *R. prolixus* [28,34,54]. Brindley’s gland outlets were funnel-like regressions, sheltered by a meshwork of cuticular microsculpture that decorated the surface (Figure 4a,b). The meshwork areas can act as evaporatoria, increasing the dispersing surface of the glandular secretion. Meshwork elements consisted of central plates (called “*chapeaux*” by Carayon [55]) with 3–5 holes connected to the adjacent ones by cuticular bridges forming the meshwork pattern (Figure 4a). Cuticular bridges delimited depressed areas called trabeculae [55]. A minutely decorated cuticular process extended along each metanotum side. Carayon [55] suggested a role for such a process called “*gouttière*” (Figure 4b). In ZR, the gutter started from the abdominal finger (Figure 4b: 21), ending in the mesothoracic flap (Figure 4b: 10). Layers and particles of solid residues of secretion (Figure 4c,d: 20) suggested that the mesothoracic flap and the metanotum gutter are evaporatoria for Brindley’s glands or other thoracic glands of ZR. Indeed, we found another outlet with a meshwork-like area below the mesothoracic flap, between the meta- and mesothorax membrane, like Brindley’s gland outlets (Figure 4c: 13). In addition, the meta–mesothoracic outlets presented organic material, suggesting secretory activity. However, the structure, anatomy, and physiology of such a putative ZR meso–metathoracic gland were not further analysed here.

### 3.2. VOCs Produced by ZR and PS

We detected 86 VOCs across all experiments (Appendix A). ZR alone produced 35 VOCs (Appendix A), while PS alone only produced 13 VOCs (Appendix A). Fifty VOCs existed in the interaction between ZR and PS (Appendix A). ZR alone made only three major VOCs (8.6%). The remaining 32 VOCs (91.4%) were present in concentrations below 100,000 ppm.

PS alone did not produce major VOCs. 2,3-Butanedione (ca. 91,000 ppm) was the most abundant PS VOC. The ZR + PS interaction included 24% major VOCs. Therefore, the largest part of the volatilome comprised minor VOCs (76%). 3-Methyl-butanal, 2-methyl-1-methylene-3-(1-methyl-ethenyl)-cyclopentane, and propyl-cyclohexane were present in the individual volatilomes of both ZR and PS (Figure 7). However, two VOCs could be in the individual PS and the predator–prey interaction volatilome (Figure 7), namely, 2,3-butenedione and 2,4,6-trimethyl-benzaldehyde. Furthermore, 2,4,6-trimethyl-benzaldehyde was the only compound constantly found in all replicates of PS alone and the PS interacting with ZR.

### 3.3. Volatilome of ZR and PS Interaction

Seven VOCs were found in the ZR + PS interaction and ZR alone (Figure 7 and Table 2). No VOC was present in all three treatments. Statistical analysis and olfactometry tests were performed on VOCs present in at least four out of seven replicates. Therefore, we only considered 2-methyl-butanoic acid, 2-methyl-propanoic acid, and 3-methyl-1-butanol because we detected all three in ZR and ZR + PS interactions (Table 2).

The multivariate GLM revealed significant differences in the volatilome between ZR and ZR + PS (*p*-value = 0.01). Furthermore, PERMANOVA also showed a significant difference (*p*-value < 0.05). SIMPER analysis indicated that the variable contributing most to the differences between ZR and ZR + PS was 2-methyl-propanoic acid, with 61.7%, followed by 2-methyl-butanoic acid, with 35%. However, only the contribution of 2-methyl-propanoic acid was significant (*p*-value = 0.03). When ZR interacted with PS, the amounts of 2-methyl-butanoic acid and 2-methyl-propanoic acid decreased (Figure 8). In contrast, 3-methyl-1-butanol increased (Figure 8). However, there was no significant change between the three VOCs produced by ZR alone and ZR + PS (GLM, *p*-value > 0.05) (Figure 8).

### 3.4. Volatilome of Philaenus Aggregation

The volatilome linked to the progressive aggregation of PS revealed ten VOCs (Appendix A). All were minor VOCs (<100,000 ppm). VOC production was higher when PS was alone (six VOCs) than during intraspecific interaction with two or more (up to ten) conspecifics (3–4 VOCs) (Appendix A). 2,4,6-Trimethyl-benzaldehyde was the only VOC always present regardless of the number of PS interactions, with an average concentration below 20,000 ppm.

### 3.5. ZR Response to VOCs Selected from Prey Interaction

2-Methyl-propanoic acid was the final choice for *Z. renardii* in 30% of olfactometer tests. The movement was 27.5% for 2-methyl-butanoic acid and 17.5% for 3-methyl-1-butanol. All differences between the choice of control vs. a VOC were highly significant (*p*-value < 0.05; *p*-value < 0.001) (Figure 9).

The average number of approaches (movements towards a given stimulus) per test to 2-methyl-propanoic acid was 0.4 ± 0.59 vs. 1.35 ± 1.23 to the control (Figure 10A). The GLMM showed significant differences between treatments (*p*-value < 0.001), whereas there were no significant differences between experiments (*p*-value = 0.06). Similarly, the average number of approaches per test was significantly lower in the case of 2-methyl-butanoic acid (0.4 ± 0.49) than in the case of departures (1.2 ± 1.01) (*p*-value < 0.001) (Figure 10B). The average number of approaches per test to 3-methyl-1-butanol was also significantly lower (0.32 ± 0.52) compared to the departures (1.47 ± 0.84) (*p*-value < 0.001) (Figure 10C). No significant differences existed between experiments (*p*-value = 0.95).

## 4. Discussion

A pair of Brindley’s glands are found in each ZR individual, male or female. These glands are in the abdomen, before the second spiracle, opening toward the metathorax, and slightly below the first spiracle. The placement and general morphology of Brindley’s gland assemblages in ZR may refer to the “type *diastomien périadénien* (F)” of Carayon ([55] page 741). Each reservoir collects the secretion from ca. 600 exocrine units. Single unicellular glands resemble the “B-type” glandular unit described in *R. prolixus* [54]. Further scrutiny will confirm the presence of one or more kinds of secretory units that should correspond to several VOCs. Each unit possesses a duct, presumably, but details of the units are beyond the scope of this study. Our study, with a few processed specimens for light microscopy and uncoated or sputtered parts for SEM observations, minimises artifacts and provides evidence of the presence of Brindley’s glands in ZR.

*Zelus renardii* and *P. spumarius* revealed differences in VOC production. The predator ZR uses olfactory communication, having a greater quantity and variety of VOCs than PS. PS aggregation caused a reduction in the number of VOCs detected by GC/MS.

*Philaenus spumarius* has fewer antennal sensory structures than other Auchenorrhyncha species, yet these structures can perform an olfactory function [56,57]. However, reducing VOC diversity with aggregation suggests PS prefers vibrational rather than olfactory communication between conspecifics [43,44]. Despite this, male PSs respond positively to female VOCs, indicating the species’ ability to perceive likely sexual olfactory stimuli, confirming the functionality of olfactory receptors [42]. VOCs in our PS volatilome were not frequently detected, so no tests were conducted to study their role in the behavioural responses to these XF vectors. Identifying the sex pheromones of PS could lead, in future studies, to developing traps for their monitoring and management in olive orchards.

The widest variety of VOCs was found in the ZR + PS interaction, suggesting that the predator rather than the prey contributes mostly to this increase. Cimicomorpha, to which ZR belongs, possess well-developed odour glands capable of producing different semiochemicals for intra- and interspecific communication [26]. Cimicomorpha VOCs include short-chain organic acids, alcohols, short-chain aldehydes and esters, alkanes, monoterpenes, aromatic alcohols, and aldehydes [27,58]. We found many of these compounds in this ZR study.

In the ZR + PS interaction and when ZR was alone, the most abundant VOC was 2-methyl-propanoic acid. In the adults of Reduviidae Triatominae, this acid is released by Brindley’s glands, when subjected to stress or dangerous situations, independently of the individual’s sex [27,29,59,60,61,62,63,64,65]. The gland system of adults of *Zelus* ssp. (Reduviidae: Harpactorinae) includes only Brindley’s glands [30]. At high doses, 2-methyl-propanoic acid elicits alarm responses in adults and juveniles of *Triatoma infestans* (Klug, 1834) [62,63] and *R. prolixus* [64]. In contrast, this acid in low doses and mixtures with other organic acids in the blends produced by Brindley’s glands has attracted the juvenile stages of *T. infestans* [66]. 2-Methyl-butanoic acid makes up the volatilome of ZR and has been found to be secreted in Brindley’s glands and, together with its derivatives, is part of the alarm pheromone blend of *T. infestans* [27]. Therefore, the production of these organic acids suggests that they may perform similar functions in ZR bionomics.

3-Methyl-1-butanol has never been reported among the compounds released by Reduviidae and can be considered one of the precursors of 2-methyl-butanoic and 2-methyl propanoic acids. 3-Methyl-1-butanol can be a derivative of isopentenyl pyrophosphate or its isomer dimethylallyl pyrophosphate. These isomers are derivatives of mevalonic acid, formed by coupling 3-unit acetyl-coenzyme A [67]. The oxidation of 3-methyl-1-butanol, commonly known as isoamyl alcohol, gives 3-methyl-butanoic acid [68]. 2-Methyl-butanoic acid can originate from 3-methyl-1-butanol or putative precursors (isopentenyl and dimethylallyl pyrophosphate) by hydrolysis, hydrogenation, and oxidation. However, 2-methyl-butanoic acid can also come from the poly-acetate pathway. Furthermore, 2-methyl-propanoic acid (or isobutyric acid) can also be formed from 3-methyl-1-butanol by successive transformation into 3-methyl-butanoic acid (primary oxidation) and 3-methyl-2-oxobutanoic acid (secondary oxidation), before finally undergoing decarboxylation. Thus, 3-methyl-1-butanol may precede the two more abundant organic acids detected in the ZR volatilome in storage in the gland reservoir. Possible transformation into 2-methyl-butanoic and 2-methyl propanoic acids occurs by spontaneous oxidation, hydrolysis, hydrogenation, and decarboxylation occurring in the environment.

3-Methyl-1-butanol, 2-methyl-propanoic, and 2-methyl-butanoic acids individually stimulated ZR to move away to areas of the olfactometer devoid of olfactory stimuli (controls), suggesting their role as alarm pheromones, as is already known for other reduviid species [27,64]. These three substances could compose the alarm pheromone blend of *Z. renardii*, with a predominance of 2-methyl-propanoic acid. Further studies will include the estimation of Kovats’ retention indices for each VOC using authentic standards and could investigate how combinations of these compounds could influence the behaviour of the ZR.

The VOCs detected are substances with low molecular weights and high volatility, typical characteristics of compounds that stimulate defensive behaviour [69]. The characteristics of VOCs allow them to quickly reach the olfactory receptors of conspecifics and to be rapidly eliminated after a disturbance. In this way, risk communication and the defensive response of conspecifics are immediate [69].

The secretion of alarm pheromones induces behavioural changes in the conspecific that detects them [70]. These consist of a series of defensive behaviour strategies, which may include the detection of danger (defence), avoidance of the threat (escape), or deterrence through attack (fight) [69]. ZRs run away and show defensive strategies, reacting to all three VOCs tested separately. This evidence confirms that 2-methyl-propanoic acid, 2-methyl-butanoic acid, and 3-methyl-1-butanol act individually as alarm pheromones in ZR. 3-Methyl-1-butanol elicited the highest escaping behaviour, followed by 2-methyl-butanoic and 2-methyl-propanoic acids.

The concentration of volatile organic acids shows a significant standard deviation when the predator is alone, whereas it is more stable during the interaction. The minor fluctuations could be due to our experimental systems’ separation networks between prey and predator. The predation behaviour of *Zelus* includes ambushing or prey stalking [71]. The presence of stable support during the interaction with PS, which provides the predator with the opportunity to hide by ambushing the prey, could cause the ZR to perceive a reduction in stress, thereby stabilising the production of alarm pheromones.

When the predator is alone, it releases significantly higher amounts of 2-methyl-propanoic acid and 2-methyl-butanoic acid, reducing their secretion in the presence of prey. Predators can modulate the secretion of alarm pheromones, and ZR can reduce the release of alarm pheromones to optimise disguise during the ambush and increase the predation rates. Only 3-methyl-1-butanol increases during the interaction and elicits an alarm response in ZR. The presence of this compound is connected to the reduction of 2-methyl-propionic acid and 2-methyl-butanoic acid, which could be precursors.

Predators use kairomones to locate their prey in natural habitats. Kairomones emitted by prey or synomones released by host plants can attract predators or parasitoids [72,73]. Prey can, likewise, develop mechanisms to detect the predator’s presence that elicit defensive responses to escape the threat of predation [69]. In addition, the alarm pheromones of some species can induce defensive behaviour in other species sharing the same habitat [70]. The same VOCs may mediate different intra- and interspecific responses [74]. Perception of predators is a crucial adaptation for reducing predation risk and maintaining prey fitness [75].

Modulating alarm pheromone production may also mark the predation area of ZR, keeping conspecific prey competitors at a distance. Some plant pests and predators use territory marking to reduce intraspecific competition for food sources. For example, females of more than 20 species of Tephritidae (Diptera) of the genera *Ceratitis*, *Anastrepha*, *Rhagoletis*, and some *Bactrocera* [76] mark the host with pheromones at oviposition sites by dragging the surrounding area with the ovipositor [77]. Host-marking pheromones inhibit further and subsequent egg-laying, reducing competition from various larvae for the same food source and cannibalism events in favour of species fitness. Moreover, pheromone marking is also known for several predator species belonging to different insect orders [78]. During the search for prey, the larvae of some Coccinellidae and Chrysopidae deposit oviposition-deterring pheromones (ODPs) to induce conspecific females to not lay eggs near insect colonies or areas already occupied [79,80,81,82]. ODPs avoid intraspecific competition for the same food resources, favouring the suitability of predators [81].

## 5. Conclusions

*Zelus renardii* has a pair of Brindley’s glands on the second urite that release their secretion through metathoracic outlets. Brindley’s glands are present in males and females of ZR and produce alarm pheromones.

*Zelus renardii* uses its capacity to produce VOCs. *Philaenus spumarius*, the main *Xylella* vector, uses vibrational communication much more than chemical communication, having a lesser ability to secrete VOCs.

*Zelus renardii*, in situations of stress or danger, can produce a mixture of substances that act as alarm pheromones towards conspecifics. This bouquet consists of 2-methyl-propanoic acid but also 2-methyl-butanoic acid and 3-methyl-1-butanol as significant components.

The predator modulates the secretion and release of this blend depending on the presence of stress or prey. When ZR interacts with PS, it reduces the production of alarm pheromones and the possibility of being detected by its prey. Alternatively, the modulation of alarm pheromone production may help the predator mark its predation territory, displacing conspecific competitors, and, thus, reducing competition.

Future evidence of the ability of *Z. renardii* to mark its predatory area may lead to plans for massive releases of the predator to contain the *Xylella* vector population without incurring cannibalism or predatory competition.

## Figures and Tables

**Figure 1 insects-14-00520-f001:**
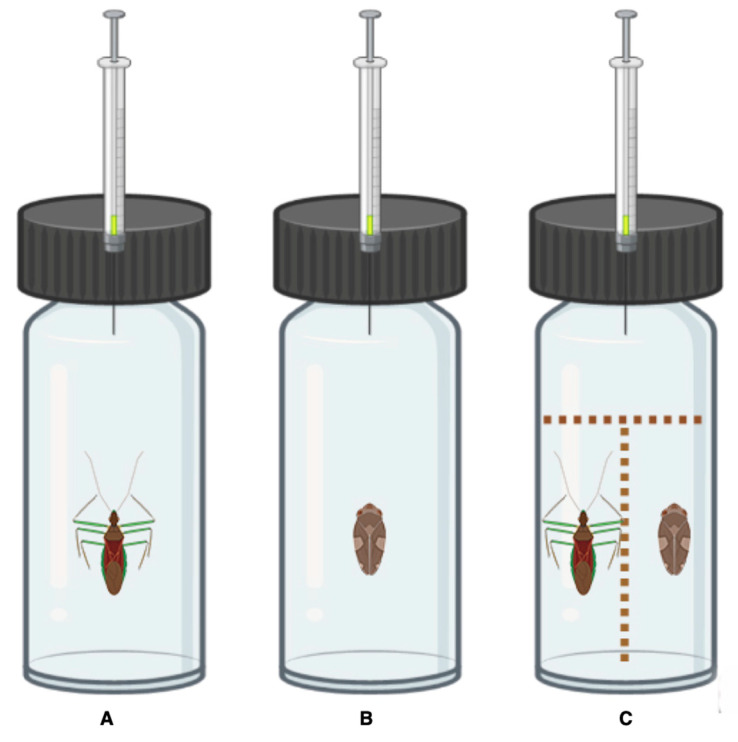
Experimental setup for volatile collection from insects. Vials: (**A**) *Zelus renardii* alone; (**B**) *Philaenus spumarius* alone; (**C**) *Z. renardii* and *P. spumarius* separated by nets (created with BioRender.com, accessed on 4 April 2023). N.B.: for pictorial fold, PS is portrayed with 3× magnification.

**Figure 2 insects-14-00520-f002:**
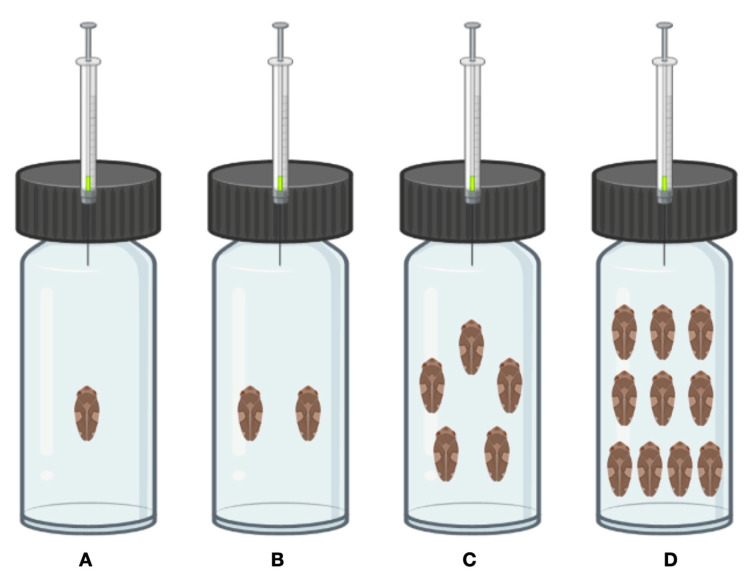
Experimental setup of *Xylella* vector VOC production. Vials: (**A**) 1 *P. spumarius*; (**B**) 2 *P. spumarius*; (**C**) 5 *P. spumarius*; (**D**) 10 *P. spumarius* (created with BioRender.com, accessed on 4 April 2023).

**Figure 3 insects-14-00520-f003:**
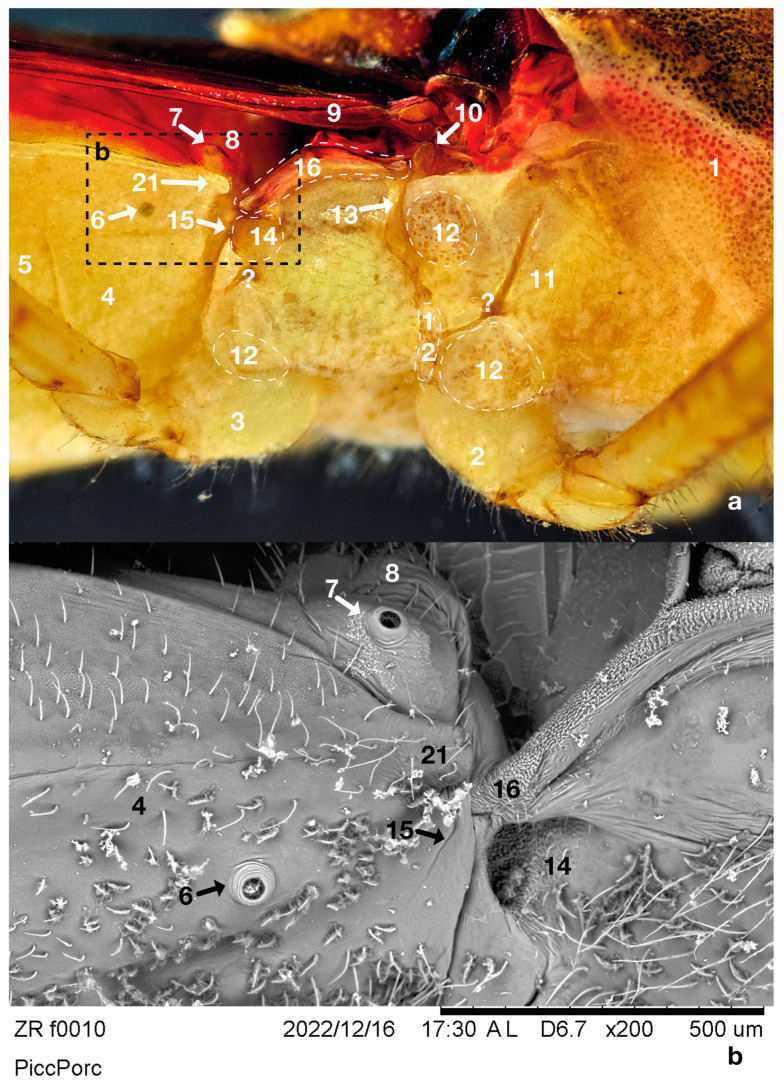
Right partial view of ZR’s thorax and abdomen: 1 = pronotum, 2 = mesothoracic coxa, 3 = metathoracic coxa, 4 = second urite, 5 = third urite, 6 = right spiracle of the second urite, 7 = right spiracle of the first urite, 8 = first urite, 9 = hemelytron, 10 = mesothoracic flap, 11 = mesopleuron, 12 = patchy cuticle areas, 13 = meshwork evaporatorium, 14 = Brindley’s gland meshwork evaporatorium, 15 = Brindley’s gland reservoir outlet place, 16 = gutter (*gouttière*), 21 = abdominal finger; (**a**) Tessovar light macroscopy; (**b**) Cryo-SEM.

**Figure 4 insects-14-00520-f004:**
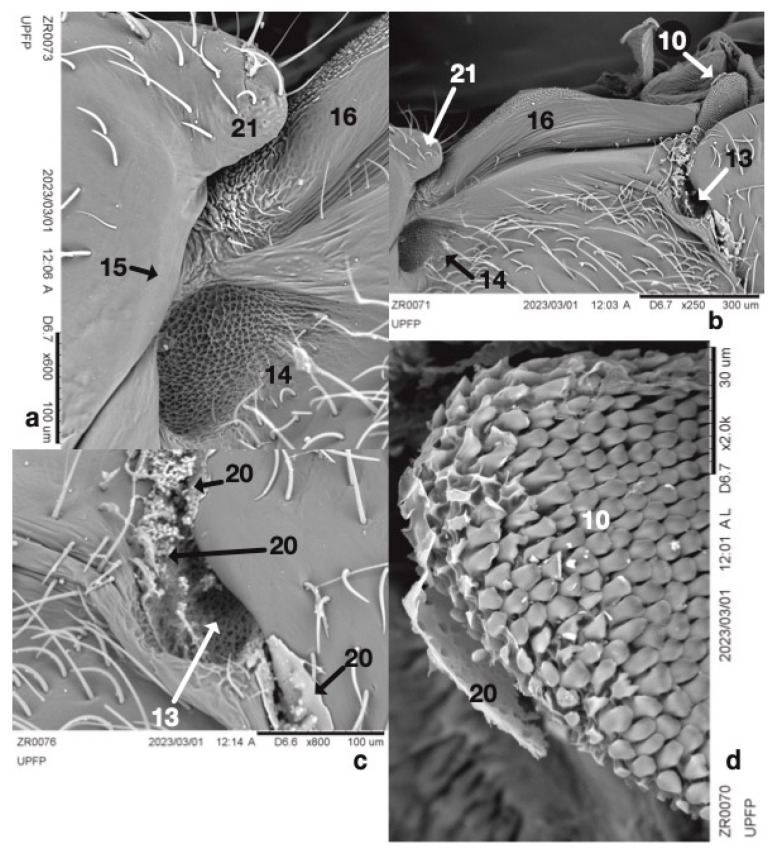
Details of right thorax and abdomen view: 10 = mesothoracic flap, 13 = meshwork evaporatorium, 14 = Brindley’s gland meshwork evaporatorium, 15 = Brindley’s gland reservoir outlet place, 16 = gutter (*gouttière*), 20 = secretions (?), 21 = abdominal finger; SEM.

**Figure 5 insects-14-00520-f005:**
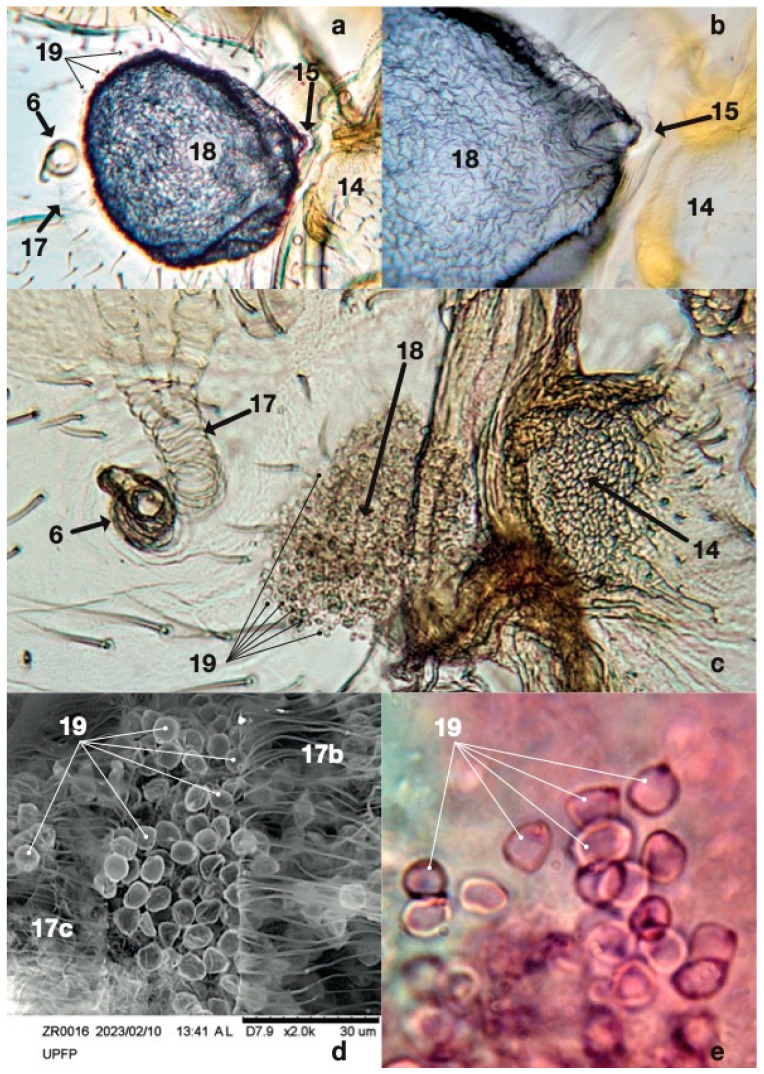
Details of Brindley’s gland: 6 = right spiracle of the second urite, 14 = Brindley’s gland meshwork evaporatorium, 15 = Brindley’s gland reservoir outlet, 17 and 17b,c = trachea and tracheal branches over Brindley’s gland, 18 = Brindley’s gland reservoir, 19 = Brindley’s gland units’ cuticles; Phomi II light microscopy, SEM.

**Figure 6 insects-14-00520-f006:**
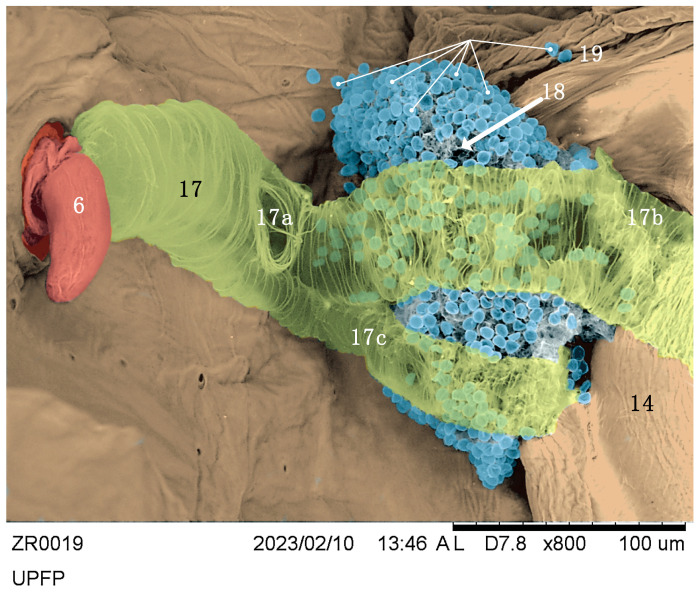
Brindley’s glands false-colour SEM: 6 = right spiracle of the second urite, 17 = trachea, 17a = sign of a broken tracheal branch, 17b, c = tracheal branches lying over Brindley’s gland, 18 = Brindley’s gland reservoir, 19 = Brindley’s gland units’ cuticles.

**Figure 7 insects-14-00520-f007:**
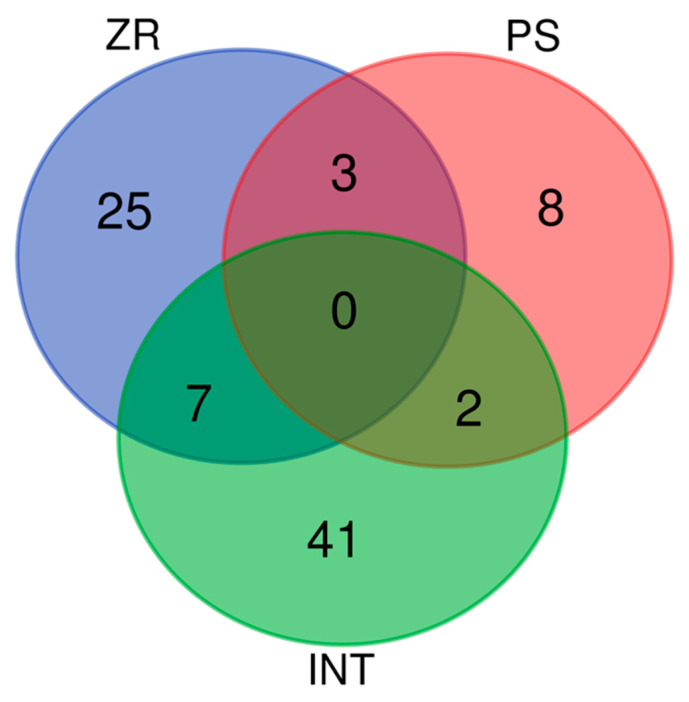
Venn diagram showing the volatiles shared between treatments. *Zelus renardii* (ZR), *Philaenus spumarius* (PS) and interaction between *Z. renardii* and *P. spumarius* (INT).

**Figure 8 insects-14-00520-f008:**
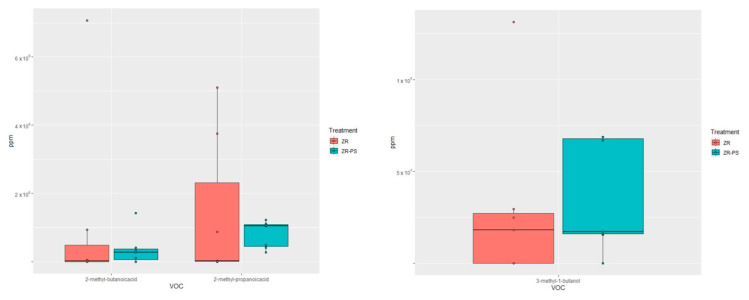
Amount of 2-methyl-butanoic acid, 2-methyl-propanoic acid, and 3-methyl-1-butanol in ZR alone (red box plots) and during ZR–PS interaction (blue box plots).

**Figure 9 insects-14-00520-f009:**
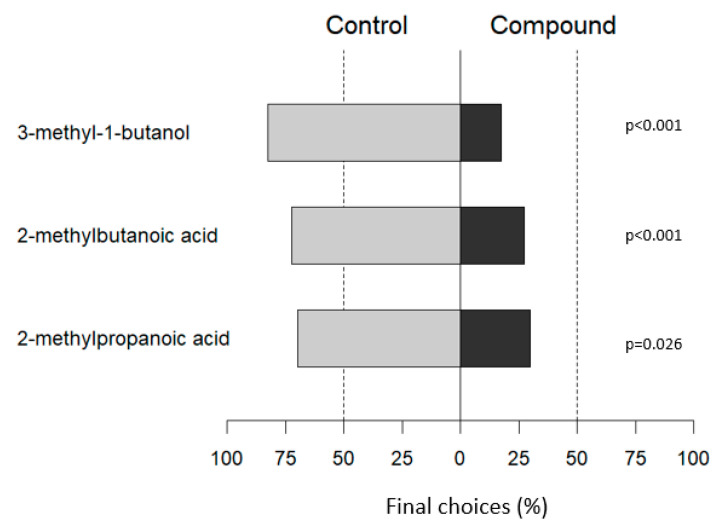
The final choice of *Z. renardii* when given a VOC (3-methyl-butanol, 2-methyl-butanoic acid, or 2-methyl-propanoic acid) or the control (empty—no VOC).

**Figure 10 insects-14-00520-f010:**
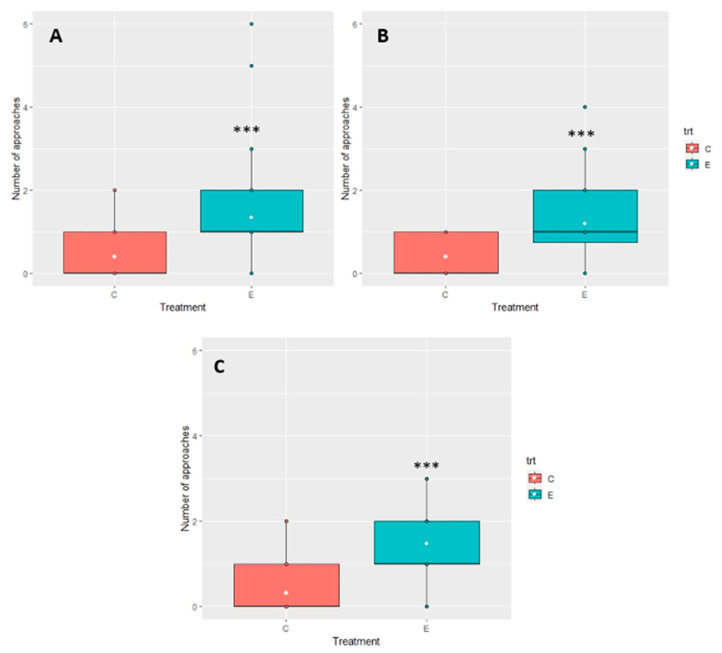
Average approaches of *Z. renardii* to VOCs: (**A**) 2-methyl-propanoic acid; (**B**) 2-methyl-butanoic acid; (**C**) 3-methyl-1-butanol. *** *p*-value < 0.001. Abbreviations: C = compound, E = empty.

**Table 1 insects-14-00520-t001:** List of *Zelus renardii* marks in pictures.

Number	Description
1	pronotum
2	mesothoracic coxa
3	metathoracic coxa
4	second urite
5	third urite
6	right spiracle of the second urite
7	right spiracle of the first urite
8	first urite
9	hemelytron
10	mesothoracic flap
11	mesopleuron
12	patchy cuticle areas
13	meshwork evaporatorium
14	Brindley’s gland meshwork evaporatorium
15	Brindley’s gland reservoir outlet place
16	gutter
17 (a–c)	trachea (and tracheal branches over Brindley’s gland)
18	Brindley’s gland reservoir
19	Brindley’s gland units’ cuticle
20	secretions (?)
21	abdominal finger

**Table 2 insects-14-00520-t002:** VOCs detected in *Z. renardii* alone and *Z. renardii*–*P. spumarius* interaction. Abbreviations: M-VOCs = major VOCs; m-VOCs = minor VOCs; R.T. = Retention Time; P.H. = peak height; No. Rep. = number of replicates for the chemical. In bold are the VOCs found in at least half of the replicates.

Compound	R.T. (min)	P.H. (ppm)	Match (%)	No. Rep.
**M-VOCs**				
**3-methyl-1-butanol**	12.656	256,956	87	5
**2-methyl-propanoic acid**	15.262	802,433	95	7
**2-methyl-butanoic acid**	18.643	507,820	68	5
2-methyl-pentanoic acid	18.741	2,106,728	72	1
**m-VOCs**				
2-pentanol	10.884	34,370	83	1
2,4,4,6-tetramethyl-hept-2-ene	24.076	50,924	50	1
1-ethyl-2-methyl-cyclohexane	24.079	54,621	50	2

## Data Availability

Data are available in the Appendix A.

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
