# Peer review of "Brindley’s Glands Volatilome of the Predator Zelus renardii Interacting with Xylella Vectors"

_insects, 2023, doi:10.3390/insects14060520_

Round 1

Reviewer 1 Report

The  English was diffficult to understand, and should be polished. 

Not very well. 

Author Response

R1: The English was difficult to understand and should be polished.

Thank you very much for the functional suggestion. As requested, we revised the English via the service provided by MDPI. We enclose the certificate of the revision.

Reviewer 2 Report

This paper describes the Brindley glands of Zelus renardii, and characterises the VOCs produced by ZR as well as the XF vector Philaneus spumarius, showing differences in VOC production when the two species are physically separated. Some very interesting findings, which could be applied to the integrated pest management to reduce the spread of XF. I have some suggestions, which I think will improve the manuscript, below.

Abstract:

There is quite a high level of detail in the abstract which might not be necessary, especially in the first paragraph. I think this could be shortened to get the key messages across. Maybe a concluding sentence would also be good to highlight what this work could mean in the wider context for e.g. integrated pest management strategies to control the spread of XF?

Line 17- Specify 'Apulia' is in Italy (e.g. Apulia, Italy)

Line 25- Be careful when discussing VOCs and semiochemicals. Authors say 'ZR also emits VOCs, however many VOCs are semiochemicals. I would remove 'ZR also emits', and say 'Zelus can secrete semiochemicals during their interaction with conspecifics and prey, including Volatile Organic Compounds....'

Line 31- Use of subjective term: 'high level' compared to what? If it's just produced by Z. renardii alone, you can remove 'high level'

Line 33- Specify that the three VOCs were presented individually to the insects

Line 40- Volatile Organic Compounds, Semiochemicals? OQDS isn't used at all in the manuscript, so maybe include in the intro/abstract, or remove it from the keywords.

Introduction

Nice overview of the current state of the literature. Remember that not all semiochemicals are VOCs- some are non-volatile- so be careful with wording in some parts of the manuscript. Quite a lot of very short paragraphs- one or two sentences- which I have suggested merging to improve the flow of the text. This could also be applied to the discussion/conclusion sections.

There's not a lot of literature on the chemical ecology aspect of the paper. Maybe the inclusion of a few more papers around Line 85 would be good here- is there much known about the chemical ecology of the insects used in this study? Known attractants/repellents?

Line 44- 'Stresses' instead of 'barriers'

Line 50- Start the sentence talking about the bacteria, as ultimately this is what is causing the losses. 'Xyella fastidiosa is a quarantine bacteria introduced from Costa Rica to Italy...'.

Line 65- I think this is repeating part of the previous paragraph, so remove. The second sentence could be moved earlier on, e.g. to line 52, just to highlight the impacts of the pathogen.

Line 70- Merge this with Line 69, I don't think this needs to be a separate paragraph.

Line 81 to line 94 can all be merged into one paragraph.

Line 97 can be merged with paragraph starting at line 95.

Materials and methods

Good overview of the methods, robust statistics, minor amendments to make. Main suggestion is about the experimental set-up for VOCs detection, and the GC-MS section.

Line 107: 'Insects collection', remove 'Camunidad Valenciana, Spain

Line 158- 'Experimental set-up for VOCs detection' instead of 'Experimental set for VOCs detection'. This section could be rearranged a bit. Chemical ecology papers tend to separate the volatile sampling tehcniques, from the GC-MS in their materials and methods section. I would therefore mention the SPME conditions in the 'Experimental set up for VOC detection', and have GC-MS as a separate section altogether. 

Line 163- remove, and at the end of the paragraph, put 'seven biological replicates were performed per treatment'

Line 164- what is a 'proper' vial? Good use of figure here.

Line 175- could a separate figure be used here to highlight this experiment? Just a suggestion. Were both males and females used in these experiments?

Line 193- more information needed for VOC identification- need to specify that these are tentative identifications, unless authentic standards were used to confirm identities?

In future, it would be worth calculating kovat's retention indices for each compound, which would require the injection of a standard mix of alkanes (C7-C22) into the GC-MS. This is an analytical tool which can provide more information as to the identity of the tentatively identified compounds. I would be happy to discuss this further with the authors.

Line 196- is there a better way of naming the VOCs, as M-VOCs and m-VOCs is a bit tricky for the reader. It might be worth not shortening, and saying 'Major VOCs' or 'minor VOCs' throughout.

Line 203- equally angled at 90 degrees? Would be good to include this

Line 205- 'The lists of VOCs per treatment (ZR, PS and ZR+PS) were compared using...' instead of 'The lists of VOCs per treatment (ZR, PS and ZR+PS) compared using...'

Line 206-'We selected the VOCs in at least four out of seven replicates for the behavioural analysis' instead of  'We selected 206 the VOCs in at least half of the replicates (four out of seven or more) for the behavioural analysis.' Correct this throughout

Line 214- have this part as a separate section at the end of the methods, called 'Chemicals'

Results

As a general rule of thumb, I think the explanation of a figure/table should come before the figure/table. I've made some suggestions below as to where this could be improved.

Line 341- I'm unsure about the total amount of VOCs produced by ZR alone. You state that 'ZR alone produces 35 VOCs... ZR alone made only three M-VOCs... the remaining 33 VOCs are present in concentrations below 100000ppm'. This equals 36 VOCs?

Line 342- 'In all our experiments', does this mean across all treatments? Including blanks? If so, I would say 'across all treatments' instead

Line 367- I think this part needs to be included with the previous sentence- its a bit disjointed at the moment. 

Line 368- remove this sentence. If it is in the GC column fitting, its non-biological, so I don't think you need to mention it

Line 388- I would move this to methods section, stating 'Forty replicates were performed for each compound'

Line 403- merge this part with the sentence at line 399. Move this section up so it comes before the figure. I think as a general rule of thumb, the figure should come after the explanation of the results for each sub-section.

Line 411- same comment as above

Line 415- I think this part fits better with the rest of the VOC data, and then put the behavioural data later on.

Discussion

Line 427- 'may refer' instead of 'may refers'

Line 436- 'Revealed differences in VOC production' might be better here

Line 440- if something is producing fewer VOCs, does that mean it prefers vibrational rather than olfactory communication?

Line 483- I don't think you need to include this information about the GC column contaminant

Line 495- It might be worth adding a sentence that these compounds were tested individually. Interesting future work could investigate how combinations of these compounds could alter insect behaviour: see Webster et al (2008) (Volatiles functioning as host cues in a blend become nonhost cues when presented alone to the black bean aphid - ScienceDirect), who showed that compounds presented individually to a bean aphid were repellent, but when combined, were attractive. 

Line 499- I think this sentence needs rewriting

Line 513- significantly higher, instead of higher. As your statistical analyses showed significant differences here, it is important to state this. This should be the case whenever trends are described (e.g. 'higher' or 'lower', if the results are significant)

Conclusion

Line 547- Is there much evidence to indicate that an organism that does not emit much VOCs therefore does not rely on chemical communication? It might be a bit speculative

Line 551- I would state 'bouquet consists of' instead of 'may consist of'. Your results show that these compounds are present in the bouquet

Line 554- Again, I would remove 'may', as your results do demonstrate that it does modulate the blend depending on the presence of prey

Very good

Author Response

Reviewer 2

R2: This paper describes the Brindley glands of Zelus renardii, and characterises the VOCs produced by ZR as well as the XF vector Philaenus spumarius, showing differences in VOC production when the two species are physically separated. Some very interesting findings, which could be applied to the integrated pest management to reduce the spread of XF. I have some suggestions, which I think will improve the manuscript, below.

Thank you very much for your appreciation.

R2: Abstract: There is quite a high level of detail in the abstract which might not be necessary, especially in the first paragraph. I think this could be shortened to get the key messages across. Maybe a concluding sentence would also be good to highlight what this work could mean in the wider context for e.g. integrated pest management strategies to control the spread of XF?

I agree with this suggestion. We streamlined the first paragraph and add a new concluding sentence.

R2: Line 17- Specify 'Apulia' is in Italy (e.g. Apulia, Italy)

Done.

R2: Line 25- Be careful when discussing VOCs and semiochemicals. Authors say 'ZR also emits VOCs, however many VOCs are semiochemicals. I would remove 'ZR also emits', and say 'Zelus can secrete semiochemicals during their interaction with conspecifics and prey, including Volatile Organic Compounds....'

Thanks. We changed the sentence, according to the suggestion.

R2: Line 31- Use of subjective term: 'high level' compared to what? If it's just produced by Z. renardii alone, you can remove 'high level'

Done.

R2: Line 33- Specify that the three VOCs were presented individually to the insects

Done.

R2: Line 40- Volatile Organic Compounds, Semiochemicals? OQDS isn't used at all in the manuscript, so maybe include in the intro/abstract, or remove it from the keywords.

Thank you very much for the advice, but we do not find it useful to repeat in the keywords terms already present in the title, summary and abstract, because they will be found anyway. We prefer to extend the possibility of being found with a parallel search by argument/topics.

R2: Introduction: Nice overview of the current state of the literature. Remember that not all semiochemicals are VOCs- some are non-volatile- so be careful with wording in some parts of the manuscript. Quite a lot of very short paragraphs- one or two sentences- which I have suggested merging to improve the flow of the text. This could also be applied to the discussion/conclusion sections.

Thank you very much for your comment. We have merged paragraphs to improve the text readability.

R2: There's not a lot of literature on the chemical ecology aspect of the paper. Maybe the inclusion of a few more papers around Line 85 would be good here- is there much known about the chemical ecology of the insects used in this study? Known attractants/repellents?

There is little literature on the chemical ecology of the insects involved. The little information is only available for Philaenus spumarius and references are already included in the bibliography (e.g. Sevarika et al. 2022; Germinara et al., 2017 and Ranieri et al., 2016).

R2: Line 44- 'Stresses' instead of 'barriers'

Done.

R2: Line 50- Start the sentence talking about the bacteria, as ultimately this is what is causing the losses. 'Xylella fastidiosa is a quarantine bacteria introduced from Costa Rica to Italy...'.

We agree with your suggestion accepting that OQDS/CoDiRO is the lethal disease killing olive trees in recent outbreak. Xylella fastidiosa entered Europe from Costa Rica on infected coffee plants.

R2: Line 65- I think this is repeating part of the previous paragraph, so remove. The second sentence could be moved earlier on, e.g. to line 52, just to highlight the impacts of the pathogen.

Thank you. We moved and removed the sentences following your suggestions.

R2: Line 70- Merge this with Line 69, I don't think this needs to be a separate paragraph.

Done.

R2: Line 81 to line 94 can all be merged into one paragraph.

Done

R2: Line 97 can be merged with paragraph starting at line 95.

Done

R2: Materials and methods: Good overview of the methods, robust statistics, minor amendments to make. Main suggestion is about the experimental set-up for VOCs detection, and the GC-MS section.

Thank you very much for your appreciation.

R2: Line 107: 'Insects collection', remove 'Camunidad Valenciana, Spain

Done

R2: Line 158- 'Experimental set-up for VOCs detection' instead of 'Experimental set for VOCs detection'. This section could be rearranged a bit. Chemical ecology papers tend to separate the volatile sampling techniques, from the GC-MS in their materials and methods section. I would therefore mention the SPME conditions in the 'Experimental set up for VOC detection', and have GC-MS as a separate section altogether.

SPME conditions are separate from GC-MS. We have slightly modified the assortment of sections to emphasise this separation. Thank you for your suggestions.

R2: Line 163- remove, and at the end of the paragraph, put 'seven biological replicates were performed per treatment'

Done

R2: Line 164- what is a 'proper' vial? Good use of figure here.

Done

R2: Line 175- could a separate figure be used here to highlight this experiment? Just a suggestion. Were both males and females used in these experiments?

Thanks! We added a new figure. We used Ps males and females and added this missing information.

R2: Line 193- more information needed for VOC identification- need to specify that these are tentative identifications, unless authentic standards were used to confirm identities? In future, it would be worth calculating kovat's retention indices for each compound, which would require the injection of a standard mix of alkanes (C7-C22) into the GC-MS. This is an analytical tool which can provide more information as to the identity of the tentatively identified compounds. I would be happy to discuss this further with the authors.

Thank you very much. We will add include in discussion/conclusion opening the opportunity for a next study.

R2: Line 196- is there a better way of naming the VOCs, as M-VOCs and m-VOCs is a bit tricky for the reader. It might be worth not shortening, and saying 'Major VOCs' or 'minor VOCs' throughout.

Done

R2: Line 203- equally angled at 90 degrees? Would be good to include this

Done

R2: Line 205- 'The lists of VOCs per treatment (ZR, PS and ZR+PS) were compared using...' instead of 'The lists of VOCs per treatment (ZR, PS and ZR+PS) compared using...'

Done

R2: Line 206-'We selected the VOCs in at least four out of seven replicates for the behavioural analysis' instead of  'We selected 206 the VOCs in at least half of the replicates (four out of seven or more) for the behavioural analysis.' Correct this throughout

Done

R2: Line 214- have this part as a separate section at the end of the methods, called 'Chemicals'

Done

R2: Results: As a general rule of thumb, I think the explanation of a figure/table should come before the figure/table. I've made some suggestions below as to where this could be improved.

Thank you so much for the advice.

R2: Line 341- I'm unsure about the total amount of VOCs produced by ZR alone. You state that 'ZR alone produces 35 VOCs... ZR alone made only three M-VOCs... the remaining 33 VOCs are present in concentrations below 100000ppm'. This equals 36 VOCs?

“total amount” do you mean the total number of VOCs?

It is a typo.

R2: Line 342- 'In all our experiments', does this mean across all treatments? Including blanks? If so, I would say 'across all treatments' instead

Done

R2: Line 367- I think this part needs to be included with the previous sentence- its a bit disjointed at the moment.

Done

R2: Line 368- remove this sentence. If it is in the GC column fitting, its non-biological, so I don't think you need to mention it

Done

R2: Line 388- I would move this to methods section, stating 'Forty replicates were performed for each compound'

Thank you so much.

R2: Line 403- merge this part with the sentence at line 399. Move this section up so it comes before the figure. I think as a general rule of thumb, the figure should come after the explanation of the results for each sub-section.

Done

R2: Line 411- same comment as above

Done

R2: Line 415- I think this part fits better with the rest of the VOC data, and then put the behavioural data later on.

Done

R2: Discussion: Line 427- 'may refer' instead of 'may refers'

Done

R2: Line 436- 'Revealed differences in VOC production' might be better here

Done

R2: Line 440- if something is producing fewer VOCs, does that mean it prefers vibrational rather than olfactory communication?

It does not necessarily prefer only vibrational communication but may prefer different means of communication in latu sensu.

R2: Line 483- I don't think you need to include this information about the GC column contaminant

Thanks.

R2: Line 495- It might be worth adding a sentence that these compounds were tested individually. Interesting future work could investigate how combinations of these compounds could alter insect behaviour: see Webster et al (2008) (Volatiles functioning as host cues in a blend become nonhost cues when presented alone to the black bean aphid - ScienceDirect), who showed that compounds presented individually to a bean aphid were repellent, but when combined, were attractive.

Done. Thanks for your advice.

R2: Line 499- I think this sentence needs rewriting

Done

R2: Line 513- significantly higher, instead of higher. As your statistical analyses showed significant differences here, it is important to state this. This should be the case whenever trends are described (e.g. 'higher' or 'lower', if the results are significant)

Done

R2: Conclusion: Line 547- Is there much evidence to indicate that an organism that does not emit much VOCs therefore does not rely on chemical communication? It might be a bit speculative

ok

R2: Line 551- I would state 'bouquet consists of' instead of 'may consist of'. Your results show that these compounds are present in the bouquet

Done

R2: Line 554- Again, I would remove 'may', as your results do demonstrate that it does modulate the blend depending on the presence of prey

Done

Reviewer 3 Report

The manuscript discusses interactions between a recently invasive insect pest, Zelus renardi, with a spittlebug, Philaenus spumarius, that has become a carrier of a plant pathogen, Xylella fastidiosa, to which olive trees are susceptible. It addresses questions that may contribute to control measures that protect olive trees against the pathogen. The reviewer has the following comments on the manuscript:

Abstract:  We consider a foreigner an invasive entity when upon acclimatisation, establishes negative interactions” should be “We consider it to have become an invasive species if it establishes negative interactions after acclimatization.”

 This reduviid is a stenophagous promising Xylella-vectors predator” should be “This reduviid is a promising stenophagous predator of Xylella vectors.”

“during juvenile steps” should be “during juvenile stages. Management then focuses on eliminating surviving adults to prevent XF infection events.”

The reviewer recommends removal of the last sentence from the abstract, although it could be suggested in the discussion. Rather, a sentence could be “Potential effects of the VOC secretions on the interaction of Z. renardi with P. spumarius are discussed”

Remainder of article

“In total 18 ZR adults were obtained for the study.”

The reviewer is confused by “Insect parts were not coated and imaged” Do you mean “insect parts were next coated and imaged” ?

“Experimental setup for VOC detection”
“versus a vial with nets”

“gave the thesis” probably should be “were dispensed”

Each VOC entered the test with four adult males of ZR, the only collected in nature” probably should be “Each VOC was tested against four adult wild-collected ZR males. Each VOC was placed ”

“Thus, the possibility that 3-methyl-1-butanol could be the precursor of the two majority organic acids detected in the ZR volatilome suggests its initial release via the ZR's Brindley glands.” This sentence is somewhat confusing. Do the authors mean “its initial production and then its conversion to the other organic acids through primary and secondary oxidation in the ZR’s”

The reviewer included comments to improve understandability

Author Response

Reviewer 3

R3: The manuscript discusses interactions between a recently invasive insect pest, Zelus renardi, with a spittlebug, Philaenus spumarius, that has become a carrier of a plant pathogen, Xylella fastidiosa, to which olive trees are susceptible. It addresses questions that may contribute to control measures that protect olive trees against the pathogen. The reviewer has the following comments on the manuscript:

Z. renardii is not an invasive and not a pest, consequently. It is an alien useful stenophagous predator. A part the common name of Leafhopper Assassin Bug indicate the predator prey preferences as reported in: https://www.mdpi.com/2075-4450/13/2/158.

R3: Abstract:“ We consider a foreigner an invasive entity when upon acclimatisation, establishes negative interactions” should be “We consider it to have become an invasive species if it establishes negative interactions after acclimatization.”

Done

R3: “This reduviid is a stenophagous promising Xylella-vectors predator” should be “This reduviid is a promising stenophagous predator of Xylella vectors.”

Done

R3: “during juvenile steps” should be “during juvenile stages. Management then focuses on eliminating surviving adults to prevent XF infection events.”

Done

R3: The reviewer recommends removal of the last sentence from the abstract, although it could be suggested in the discussion. Rather, a sentence could be “Potential effects of the VOC secretions on the interaction of Z. renardiwith P. spumarius are discussed”

Done

R3: “In total 18 ZR adults were obtained for the study.”

We have changed. Thanks for the advice.

R3: The reviewer is confused by “Insect parts were not coated and imaged” Do you mean “insect parts were next coated and imaged” ?

We modify this sentence to better explain the SEM imaging.

R3: “Experimental setup for VOC detection”

Done

R3: “versus a vial with nets”

Done

R3: “gave the thesis” probably should be “were dispensed”

Done

R3: “Each VOC entered the test with four adult males of ZR, the only collected in nature” probably should be “Each VOC was tested against four adult wild-collected ZR males. Each VOC was placed ”

Done

R3: “Thus, the possibility that 3-methyl-1-butanol could be the precursor of the two majority organic acids detected in the ZR volatilome suggests its initial release via the ZR's Brindley glands.” This sentence is somewhat confusing. Do the authors mean “its initial production and then its conversion to the other organic acids through primary and secondary oxidation in the ZR’s”

Solved.

Round 2

Reviewer 2 Report

I have been through the paper, and through the individual comments in the first round of reviews, and I am satisfied with the changes made. Congratulations to the authors. If they would like to talk about Kovats retention indices, then I would be happy to discuss further with them.